# Forward Super-Resolution: How Can GANs Learn Hierarchical Generative Models for Real-World Distributions

**Zeyuan Allen-Zhu**
Allen-Zhu Research
`zeyuan2023@allen-zhu.com`

**Yuanzhi Li**
Mohamed bin Zayed University of AI
`Yuanzhi.Li@mbzuai.ac.ae`

## Abstract

Generative adversarial networks (GANs) are among the most successful models for learning high-complexity, real-world distributions. However, in theory, due to the highly non-convex, non-concave landscape of the minmax training objective, GAN remains one of the least understood deep learning models. In this work, we formally study how GANs can efficiently learn certain hierarchically generated distributions that are close to the distribution of real-life images. We prove that when a distribution has a structure that we refer to as *forward super-resolution*, then simply training generative adversarial networks using stochastic gradient descent ascent (SGDA) can learn this distribution efficiently, both in sample and time complexities. We also provide empirical evidence that our assumption "forward super-resolution" is very natural in practice, and the underlying learning mechanisms that we study in this paper (to allow us efficiently train GAN via SGDA in theory) simulates the actual learning process of GANs on real-world problems. [1]

## 1 Introduction

Generative adversarial networks (GANs) (Goodfellow et al., 2014) are among the successful models for learning high-complexity, real-world distributions. In practice, by training a *min-max* objective with respect to a generator and a discriminator consisting of multi-layer neural networks, using simple local search algorithms such as stochastic gradient descent ascent (SGDA), the *generator* can be trained *efficiently* to generate samples from complicated distributions (such as the distribution of images). But, from a theoretical perspective, how can GANs learn these distributions efficiently given that learning much simpler ones are already computationally hard (Chen et al., 2022a)?

Answering this in full can be challenging. However, following the tradition of learning theory, one may hope for discovering some concept class consisting of non-trivial target distributions, and showing that using SGDA on a min-max generator-discriminator objective, not only the training converges in poly-time (a.k.a. trainability), but more importantly, the generator learns the target distribution to good accuracy (a.k.a. learnability). To this extent, we believe prior theory works studying GANs may still be somewhat inadequate.

- Some existing theories focus on properties of GANs at the *global-optimum* (Arora et al., 2017; 2018; Bai et al., 2018; Unterthiner et al., 2017); while it remains unclear how the training process can find such global optimum efficiently.

- Some theories focus on the trainability of GANs, in the case when the loss function is convex-concave (so a global optimum can be reached), or when the goal is only to find a critical point (Daskalakis & Panageas, 2018a;b; Gidel et al., 2018; Heusel et al., 2017; Liang & Stokes, 2018; Lin et al., 2019; Mescheder et al., 2017; Mokhtari et al., 2019; Nagarajan & Kolter, 2017). Due to non-linear neural networks used in practical GANs, it is highly unlikely that the min-max training objective is convex-concave. Also, it is unclear whether such critical points correspond to learning certain non-trivial distributions (like image distributions).

---

[1] Full version of this paper can be found on `https://arxiv.org/abs/2106.02619`.

- Even if the generator and the discriminator are linear functions over prescribed feature mappings — such as the neural tangent kernel (NTK) feature mappings — see (Allen-Zhu et al., 2019b; Arora et al., 2019; Daniely et al., 2016; Du et al., 2018; Jacot et al., 2018; Zou et al., 2018) and the references therein — the training objective can still be non-convex-concave.
- Some other works introduced notions such as proximal equilibria (Farnia & Ozdaglar, 2020) or added gradient penalty (Mescheder et al., 2018) to improve training convergence. Once again, they do not study the "learnability" aspect of GANs. In particular, Chen et al. (2022b) even explicitly argue that min-max optimality may not directly imply distributional learning for GANs.
- Even worse, unlike supervised learning where some non-convex learning problems can be shown to haveno bad local minima (Ge et al., 2016), to the best of our knowledge, it still remains unclear what the qualities are of those critical points in GANs except in the most simple setting when the generator is a one-layer neural network (Feizi et al., 2017; Lei et al., 2019).

(We discuss some other related works in distributional learning in the full version.)

Motivate by this *huge gap* between theory and practice, in this work, we make a preliminary step by showing that, when an image-like distribution is hierarchically generated (using an unknown $O(1)$-layered target generator) with a structural property that we refer to as *forward super-resolution*, then under certain mild regularity conditions, such distribution can be *efficiently* learned — both in sample and time complexity — by applying SGDA on a GAN objective.[2] Moreover, to justify the scope of our theorem, we provide empirical evidence that forward super-resolution *holds for practical image distributions*, and most of our regularity conditions hold in practice as well.

We believe our work extends the scope of traditional distribution learning theory to the regime of learning continuous, complicated real-world distributions such as the distribution of images, which are often generated through some *hierarchical generative models*. We draw connections between traditional distribution learning techniques such as method of moments to the generator-discriminator framework in GANs, and shed lights on what GANs are doing beyond these techniques.

### 1.1 FORWARD SUPER-RESOLUTION: A SPECIAL PROPERTY OF IMAGES

Real images can be viewed in multiple resolutions without losing the semantics. In other words, the resolution of an image can be greatly reduced (e.g. by taking the average of nearby pixels), while still keeping the structure of the image. Motivated by this observation, the seminal work of Karras et al. (2018) proposes to train a generator progressively: the lower levels of the generator are trained first to generate the lower-resolution version of images, and then the higher levels are gradually trained to generate higher and higher resolution images. In our work, we formulate this property of images as what we call *forward super-resolution*:

> **Forward super-resolution property (mathematical statement see Section 2.1):**
>
> There exists a generator $G$ as an $L$-hidden-layer neural network with ReLU activation, where each $G_\ell$ represent the hidden neuron values at layer $\ell$, and there exists matrices $\mathbf{W}_\ell$ such that
>
> $$\text{the distribution of images at resolution level } \ell \text{ is given by } \mathbf{W}_\ell G_\ell$$
>
> and the randomness is taken over the randomness of the input to $G$ (usually standard Gaussian).

In plain words, we assume there is an (unknown) neural network $G$ whose hidden layer $G_\ell$ can be used to generate images of resolution level $\ell$ (larger $\ell$ means better resolution) via a linear transformation, typically a deconvolution. We illustrate that this assumption holds on practical GAN training in Figure 1. This assumption is also made in the practical work (Karras et al., 2018). Moreover, there is a body of works that directly use GANs or deconvolution networks for super-resolution (Bulat & Tzimiropoulos, 2018; Ledig et al., 2017; Lim et al., 2017; Wang et al., 2018; Zhang et al., 2018).

## 2 PROBLEM SETUP

Throughout this paper, we use $a = \text{poly}(b)$ for $a > 0, b > 1$ to denote that there are absolute constants $C_1 > C_2 > 0$ such that $b^{C_2} < a < b^{C_1}$. For a target learning error $\varepsilon \in [\frac{1}{d^{\omega(1)}}, \frac{1}{\text{poly}(d)}]$,

---

[2]Plus a simple SVD warmup initialization that is easily computable from the covariance of image patches.

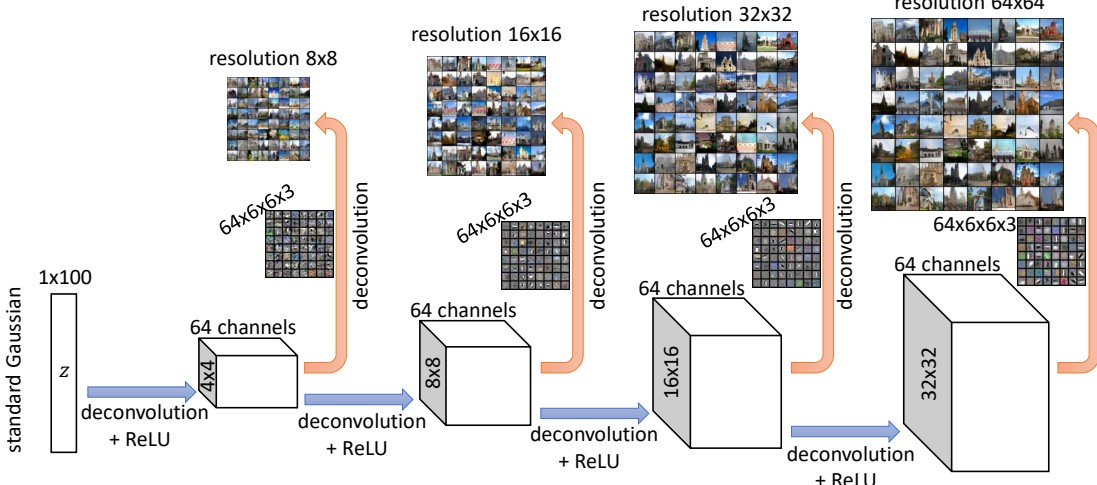

Figure 1: Illustration of the **forward super-resolution structure**. Church images generated by 4-hidden-layer deconvolution network (DCGAN), trained on LSUN Church data set using multi-scaled gradient (Karnewar & Wang, 2019). The structure of the generator is shown as above, and there is a ReLU activation between each layers. We use simple average pooling to construct low resolution images from the original training images.

we use "w.h.p." to indicate with probability $\geq 1 - \frac{1}{(d/\varepsilon)^{\omega(1)}}$. Recall $\mathsf{ReLU}(z) = \max\{z, 0\}$. In this paper, for theoretical purpose we consider a smoothed version $\widetilde{\mathsf{ReLU}}(z)$ and a leaky version $\mathsf{LeakyReLU}(z)$. We give their details in the full version, and they are different from $\mathsf{ReLU}(z)$ only by a sufficiently small quantity $1/\mathsf{poly}(d/\varepsilon)$.

## 2.1 THE TARGET DISTRIBUTION: FORWARD SUPER-RESOLUTION STRUCTURE

We consider outputs (think of them as images) $\{X_\ell^\star\}_{\ell \in [L]}$, where $X_L^\star$ is the final output, and $X_\ell^\star$ is the "low resolution" version of $X_L^\star$, with $X_1^\star$ having the lowest resolution. We think of each $\ell$-resolution image $X_\ell^\star$ consists of $d_\ell$ patches (for example, an image of size $36 \times 36$ contains 36 patches of size $6 \times 6$), where $X_\ell^\star = (X_{\ell,j}^\star)_{j \in [d_\ell]}$ and each $X_{\ell,j}^\star \in \mathbb{R}^d$. Typically, such "resolution reduction" from $X_L^\star$ to $X_\ell^\star$ can be given by sub-sampling, average pooling, Laplacian smoothing, etc., but we do not consider any specific form of resolution reduction in this work, as it does not matter for our main result to hold.

Formally, we define the **forward super-resolution** property as follows. We are given samples of the form $G^\star(z) = (X_1^\star, X_2^\star, \cdots, X_L^\star)$, where each $X_\ell^\star$ is generated by an **unknown** target neural network $G^\star(z)$ at layer $\ell$, with respect to a standard Gaussian $z \sim \mathcal{N}(0, \mathbf{I}_{m_0 \times m_0})$.

- The basic resolution: for every $j \in [d_1]$,

$$X_{1,j}^\star = \mathbf{W}_{1,j}^\star \mathcal{S}_{1,j}^\star \in \mathbb{R}^d \quad \text{for} \quad \mathcal{S}_{1,j}^\star = \mathcal{S}_{1,j}^\star(z) = \mathsf{ReLU}(\mathbf{V}_{1,j}^\star z - b_{1,j}^\star) \in \mathbb{R}_{\geq 0}^{m_1}$$

where $\mathbf{V}_{1,j}^\star \in \mathbb{R}^{m_1 \times m_0}$, $b_{1,j}^\star \in \mathbb{R}^{m_1}$ and we assume $\mathbf{W}_{1,j}^\star \in \mathbb{R}^{d \times m_1}$ is column orthonormal.

- For every $\ell > 1$, the image patches at resolution level $\ell$ are given as: for every $j \in [d_\ell]$,

$$X_{\ell,j}^\star = \mathbf{W}_{\ell,j}^\star \mathcal{S}_{\ell,j}^\star \in \mathbb{R}^d \quad \text{for} \quad \mathcal{S}_{\ell,j}^\star = \mathsf{ReLU}\left(\sum_{j' \in \mathcal{P}_{\ell,j}} \mathbf{V}_{\ell,j,j'}^\star \mathcal{S}_{\ell-1,j'}^\star - b_{\ell,j}^\star\right) \in \mathbb{R}_{\geq 0}^{m_\ell}$$

where $\mathbf{V}_{\ell,j,j'}^\star \in \mathbb{R}^{m_\ell \times m_{\ell-1}}$, $b_{\ell,j}^\star \in \mathbb{R}^{m_\ell}$, and we assume $\mathbf{W}_{\ell,j}^\star \in \mathbb{R}^{d \times m_\ell}$ is column orthonormal. Here, $\mathcal{P}_{\ell,j} \subseteq [d_{\ell-1}]$ can be any subset of $[d_{\ell-1}]$ to describe the connection graph.

**Remark.** For every layer $\ell$, $j \in [d_\ell]$, $r \in [m_\ell]$, one should view of each $[\mathcal{S}_{\ell,j}^\star]_r$ as the *r-th channel in the j-th patch at layer $\ell$*. One should think of $\sum_{j' \in \mathcal{P}_{\ell,j}} \mathbf{V}_{\ell,j,j'}^\star \mathcal{S}_{\ell-1,j'}^\star$ as the linear "deconvolution" operation over hidden layers. When the network is a deconvolutional network such as in DCGAN (Radford et al., 2015), we have all $\mathbf{W}_{\ell,j}^\star = \mathbf{W}_\ell^\star$; but we do not restrict ourselves to this case. As illustrated in Figure 2, we should view $\mathbf{W}_{\ell,j}^\star$ as a matrix consisting of the "edge-color" features to generate image patches. Crucially, when we get a data sample $G^\star(z) = (X_1^\star, X_2^\star, \cdots, X_L^\star)$,

the learning algorithm **does not know** the underlying $z$ used for this sample.

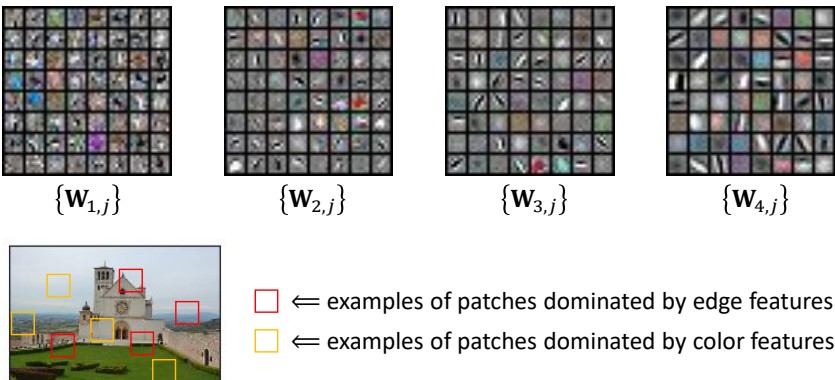

$$\{\mathbf{W}_{1,j}\} \qquad \{\mathbf{W}_{2,j}\} \qquad \{\mathbf{W}_{3,j}\} \qquad \{\mathbf{W}_{4,j}\}$$

□ ⇐ examples of patches dominated by edge features

□ ⇐ examples of patches dominated by color features

Figure 2: Visualization of the ***edge-color features*** learned in the output layers of $G^\star$. Each $\mathbf{W}_{\ell,j}$ is of dimension $m_\ell \times d = 64 \times 108 = 64 \times (6 \times 6 \times 3)$. The network is trained as in Figure 1. Note: For a deconvolutional output layer, all $\mathbf{W}_{\ell,j}$'s are equal for all $j \in [m_\ell]$.

Although our analysis holds in many settings, for simplicity, in this paper we focus on the following parameter regime (for instance, $d_\ell$ can be $d^\ell$):

**Setting 2.1.** $L = O(1)$, *each* $m_\ell = \mathsf{poly}(d)$, *each* $d_\ell = \mathsf{poly}(d)$, *and each* $\|\mathbf{V}^\star_{\ell,j,j'}\|_F \leq \mathsf{poly}(d)$.

To efficient learn a distribution with the "forward super-resolution" structure, we assume that the true distribution in each layer of $G^\star$ satisfies the following "sparse coding" structure:

**Assumption 2.2** (sparse coding structure). *For every* $\ell \in [L], j \in [d_\ell], p \in [m_\ell]$, *there exists some* $k_\ell \ll m_\ell$ *with* $k_\ell \in \left[\Omega(\log m_\ell), m_\ell^{o(1)}\right]$ *such that — recalling* $\mathcal{S}^\star_{\ell,j} \geq 0$ *is a non-negative vector:*[3]

$$\mathbf{Pr}_{z \sim \mathcal{N}(0,\mathbf{I})} \left[[\mathcal{S}^\star_{\ell,j}]_p > 0\right] \leq \frac{\mathsf{poly}(k_\ell)}{m_\ell}, \quad \mathbb{E}_{z \sim \mathcal{N}(0,\mathbf{I})} \left[[\mathcal{S}^\star_{\ell,j}]_p\right] \geq \frac{1}{\mathsf{poly}(k_\ell)m_\ell}$$

$$\text{w.h.p. over } z\colon \quad \|\mathcal{S}^\star_{\ell,j}\|_\infty \leq \mathsf{poly}(k_\ell), \quad \|\mathcal{S}^\star_{\ell,j}\|_0 \leq k_\ell$$

*Moreover, we within the same patch, the channels are pair-wise and three-wise "not-too-positively correlated":* $\forall p, q, r \in [m_\ell], p \neq q \neq r$:

$$\mathbf{Pr}_z \left[[\mathcal{S}^\star_{\ell,j}]_p > 0, [\mathcal{S}^\star_{\ell,j}]_q > 0\right] \leq \varepsilon_1 = \frac{\mathsf{poly}(k_\ell)}{m_\ell^2}, \quad \mathbf{Pr}_z \left[[\mathcal{S}^\star_{\ell,j}]_p > 0, [\mathcal{S}^\star_{\ell,j}]_q > 0, [\mathcal{S}^\star_{\ell,j}]_r > 0\right] \leq \varepsilon_2 = \frac{1}{m_\ell^{2.01}}$$

*Remark* 2.3. Although we have borrowed the notion of sparse coding, our task is very different from traditional sparse coding. We discuss more in the full version.

**Sparse coding structure in practice.** The sparse coding structure is very natural in practice for generating images (Gu et al., 2015; Zheng et al., 2010). As illustrated in Figure 2, typically, after training, the output layer of the generator network $\mathbf{W}_{\ell,j}$ forms edge-color features. It is known that such edge-color features are indeed a (nearly orthogonal) basis for images, under which the coefficients are indeed *very sparse*. We refer to (Allen-Zhu & Li, 2021) for concrete measurement of the sparsity and orthogonality. The "not-too-positive correlation" property is also very natural: for instance, in an image patch if an edge feature is used, it is less likely that a color feature shall be used (see Figure 2). In Figure 3, we demonstrate that for some learned generator networks, the activations indeed become sparse and "not-too-positively correlated" after training.

Crucially, we have *only* assumed that channels are not-too-positively correlated *within a single patch*, and channels across different patches (e.g $\mathcal{S}^\star_{\ell,1}$ and $\mathcal{S}^\star_{\ell,2}$) can be arbitrarily dependent. This makes sure the global structure of the images can still be quite arbitrary, so Assumption 2.2 can *indeed be reasonable*.[4]

---

[3]Here, $\mathsf{poly}(k_\ell)$ can be an arbitrary polynomial such as $(k_\ell)^{100}$, and our final theorem holds for sufficiently large $d$ because $d^{o(1)} > \mathsf{poly}(k_\ell)$.

[4]Within a patch, it is natural that the activations are not-too-positively correlated: for example, once a patch chooses to use a horizontal edge feature, it is *less likely* that it will pick up another vertical edge feature.

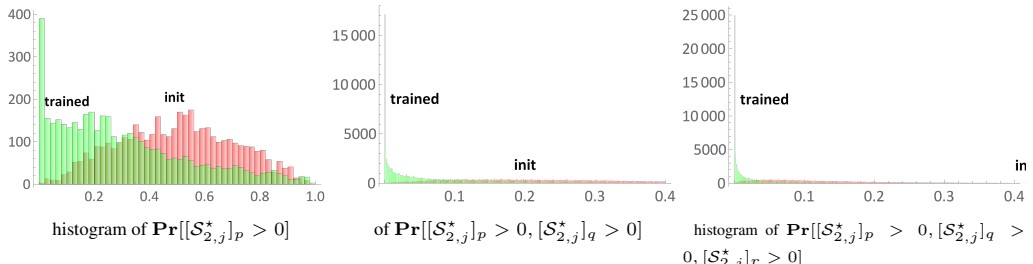


histogram of $\mathbf{Pr}[[\mathcal{S}_{2,j}^\star]_p > 0]$     of $\mathbf{Pr}[[\mathcal{S}_{2,j}^\star]_p > 0, [\mathcal{S}_{2,j}^\star]_q > 0]$     histogram of $\mathbf{Pr}[[\mathcal{S}_{2,j}^\star]_p > 0, [\mathcal{S}_{2,j}^\star]_q > 0, [\mathcal{S}_{2,j}^\star]_r > 0]$


Figure 3: Histograms at random init vs. after training for layer $\ell = 2$ of the architecture in Figure 1. Experiments for other layers can be found in Figure 6. It shows the learned network has sparse, not-too-positively correlated hidden activations (we did not regularize sparsity or correlation during training). Thus, it can be reasonable to assume that the activations of the *target network* are also sparse.

**Missing details.** We also make very mild non-degeneracy and anti-concentration assumptions, and give examples for networks satisfying our assumptions. We defer them to the full version.

## 2.2 LEARNER NETWORK (GENERATOR)

We use a learner network (generator) that has the same structure as the (unknown) target network:

- The image of the first resolution is given by:

$$X_{1,j} = \mathbf{W}_{1,j}\mathcal{S}_{1,j} \in \mathbb{R}^d \quad \text{for} \quad \mathcal{S}_{1,j} = \mathsf{LeakyReLU}(\mathbf{V}_{1,j}z - b_{1,j}) \in \mathbb{R}^{m_1}$$

  for $\mathbf{W}_{1,j} \in \mathbb{R}^{d \times m_1}$, $\mathbf{V}_{1,j} \in \mathbb{R}^{m_1 \times m_0'}$ with $m_0' \geq 2d_1 m_1$.
- The image of higher resolution is given by:

$$X_{\ell,j} = \mathbf{W}_{\ell,j}\mathcal{S}_{\ell,j} \in \mathbb{R}^d \quad \text{for} \quad \mathcal{S}_{\ell,j} = \mathsf{LeakyReLU}\left(\sum_{j' \in \mathcal{P}_{\ell,j}} \mathbf{V}_{\ell,j,j'}\mathcal{S}_{\ell-1,j'} - b_{\ell,j}\right) \in \mathbb{R}^{m_\ell}$$

  for $\mathbf{W}_{\ell,j} \in \mathbb{R}^{d \times m_\ell}$ and $\mathbf{V}_{\ell,j} \in \mathbb{R}^{m_\ell \times m_{\ell-1}}$.

One can view $\mathcal{S}_\ell$ as the $\ell$-th hidden layer. We use $G_\ell(z)$ to denote $(X_{\ell,j})_{j \in [d_L]}$. We point out both the target and the learner network we study here can be standard deconvolution networks.

## 2.3 THEOREM STATEMENT

This papers proves that by applying SGDA on a generator-discriminator objective (algorithm to be described in Section 3), we can learn the target distribution using the above generator network.

**Theorem E.1.** *For every $d > 0$, every $\varepsilon \in [\frac{1}{d^{\omega(1)}}, \frac{1}{2}]$, letting $G(z) = (X_1(z), \ldots, X_L(z))$ be the generator learned after running Algorithm 6 (which runs in time/sample complexity $\mathsf{poly}(d/\varepsilon)$), then w.h.p. there is a column orthonormal matrix $\mathbf{U} \in \mathbb{R}^{m_0 \times m_0'}$ such that*

$$\mathbf{Pr}_{z \sim \mathcal{N}(0, \mathbf{I}_{m_0' \times m_0'})}\left[\left\|G^\star(\mathbf{U}z) - G(z)\right\|_2 \leq \varepsilon\right] \geq 1 - \frac{1}{(d/\varepsilon)^{\omega(1)}} \ .$$

*In particular, this implies the 2-Wasserstein distance $\mathcal{W}_2(G(\cdot), G^\star(\cdot)) \leq \varepsilon$.*

## 3 LEARNING ALGORITHM

In this section, we define the learning algorithm using min-max optimization. We assume one access polynomially many (i.e., $\mathsf{poly}(d/\varepsilon)$) i.i.d. samples from the true distribution $X^\star = (X_1^\star, X_2^\star, \cdots, X_L^\star)$, generated by the (unknown) target network defined in Section 2.1.

To begin with, we use a simple SVD warm start to initialize (only) the output layers $\mathbf{W}_{\ell,j}$ of the network. It merely involves a simple estimator of certain truncated covariance of the data. We defer

---

We also point out that if $[\mathcal{S}_{\ell,j}^\star]_p$'s are all independent, then $\mathbf{Pr}[[\mathcal{S}_{\ell,j}^\star]_p > 0, [\mathcal{S}_{\ell,j}^\star]_q > 0] \approx \frac{1}{m_\ell^2} \leq \varepsilon_1$ and $\mathbf{Pr}[[\mathcal{S}_{\ell,j}^\star]_p > 0, [\mathcal{S}_{\ell,j}^\star]_q > 0, [\mathcal{S}_{\ell,j}^\star]_r > 0] \approx \frac{1}{m_\ell^3} \ll \varepsilon_2$.

it to the full paper. Also, we refer stochastic gradient descent ascent SGDA (on the GAN objective) to an algorithm to optimize $\min_x \max_y f(x, y)$, where the inner maximization is trained at a faster frequency. We call it Algorithm 4 and include its pseudocode in the full paper.

To make the learning process more clear, we *break the learning into multiple parts* and introduce them separately in this section:

- GAN_OutputLayer: to learn output matrices $\{\mathbf{W}_{\ell,j}\}$ per layer.
- GAN_FirstHidden: to learn hidden matrices $\{\mathbf{V}_{1,j}\}$ for the first layer.
- GAN_FowardSuperResolution: to learn higher-level hidden layers $\{\mathbf{V}_{\ell,j,j'}\}$.

We use different discriminators at different parts for our theory analysis, and shall characterize what discriminator does and how the generator can leverage the discriminator to learn the target distribution. We point out, although one can add up and mix those discriminators to make it a single one, how to use a same discriminator across the entire algorithm remains open.

At the end of this section, we shall explain how they are combined to give the final training process. *Remark* 3.1. Although we apply an SVD algorithm to get a *warm start* on the output matrices $\mathbf{W}_{\ell,j}$, the majority of the learning of $\mathbf{W}_{\ell,j}$ (e.g., to any small $\varepsilon = \frac{1}{\mathsf{poly}(d)}$ error) is still done through gradient descent ascent. We point out that the seminal work on neurally plausible dictionary learning also considers such a warm start (Arora et al., 2015a).

## 3.1 LEARN THE OUTPUT LAYER

We first introduce the discriminator for learning the output layer. For each resolution $\ell \in [L]$ and patch $j \in [d_\ell]$, we consider a one-hidden-layer discriminator

$$D_{\ell,j}^{(1)}(Y) := \sum_{r \in [m_\ell]} \left( \mathsf{ReLU}'([(\mathbf{W}_{\ell,j}^D)^\top Y_j]_r - \mathbb{b}) \langle Y_j, V_{\ell,j,r}^D \rangle \right) \ ,$$

where the input is either $Y = X_\ell^\star$ (from the true distribution) or $Y = X_\ell$ (from the generator).

Above, on the discriminator side, we have default parameter $\mathbf{W}_{\ell,j}^D, \mathbb{b}$ and trainable parameters $V_{\ell,j}^D = (V_{\ell,j,r}^D)_{r \in [m_\ell]}$ where each $V_{\ell,j,r}^D \in \mathbb{R}^d$. On the generator side, we have trainable parameters $\mathbf{W}_{\ell,j}$ (which are used to calculate $X_\ell$). (We use superscript $^D$ to emphasize $\mathbf{W}_{\ell,j}^D$ are the parameters for the discriminator, to distinguish it from $\mathbf{W}_{\ell,j}$.)

In our pseudocode GAN_OutputLayer (see Algorithm 1), for fixed $\mathbf{W}_{\ell,j}^D, \mathbb{b}$, we perform gradient descent ascent on the GAN objective with discriminator $D_{\ell,j}^{(1)}$, to minimize over $V_{\ell,j}^D$ and maximize over $\mathbf{W}_{\ell,j}$. In our final training process (to be given in full in Algorithm 6), we shall start with some $\mathbb{b} \ll 1$ and periodically decrease it; and we shall periodically set $\mathbf{W}_{\ell,j}^D = \mathbf{W}_{\ell,j}$ to be the same as the generator from a previous check point.

- Simply setting $\mathbf{W}_{\ell,j}^D = \mathbf{W}_{\ell,j}$ involves *no additional learning*, as all the learning is still being done using gradient descent ascent.
- In practice, the first hidden layer of the discriminator indeed learns the edge-color detectors (see Figure 8 in the full paper), similar to the edge-color features in the output layer of the generator. Thus, setting $\mathbf{W}_{\ell,j}^D = \mathbf{W}_{\ell,j}$ is *a reasonable approximation*. As we pointed out, how to analyze a discriminator that exactly matches practice is an important open theory direction.

INTUITION: WHAT DOES THE DISCRIMINATOR DO? To further understand the algorithm, we can see that for each $V_{\ell,j,r}^D$, when its norm is fixed, then the maximizer is obtained at

$$V_{\ell,j,r}^D \propto \left( \mathbb{E}[\mathsf{ReLU}'([(\mathbf{W}_{\ell,j}^D)^\top X_{\ell,j}^\star]_r - b) X_{\ell,j}^\star] - \mathbb{E}[\mathsf{ReLU}'([(\mathbf{W}_{\ell,j}^D)^\top X_{\ell,j}]_r - b) X_{\ell,j}] \right)$$

Thus, for the generator to further minimize the objective, the generator will learn to *match the moments of the true distribution*. In other words, generator wants to ensure

$$\mathbb{E}[\mathsf{ReLU}'([(\mathbf{W}_{\ell,j}^D)^\top X_{\ell,j}]_r - b) X_{\ell,j}] \approx \mathbb{E}[\mathsf{ReLU}'([(\mathbf{W}_{\ell,j}^D)^\top X_{\ell,j}^\star]_r - b) X_{\ell,j}^\star]$$

In this paper, we prove that such a truncated moment can be matched efficiently simply by running gradient descent ascent. Moreover, we empirically observe (see Figure 4) that *GANs can indeed do*

---

**Algorithm 1** (GAN_OutputLayer) method of moments

---

**Input:** $\mathbf{W}^{(0)}_{\ell,j}, b, \ell, j$

1: Set $\mathbf{W}^D_{\ell,j} \leftarrow \mathbf{W}^{(0)}_{\ell,j}$; $\mathrm{b} \leftarrow bm^{0.152}$; $N \leftarrow \frac{1}{\mathsf{poly}(d/\varepsilon)}, \eta \leftarrow \frac{1}{\mathsf{poly}(d/\varepsilon)}, T \leftarrow \frac{\mathsf{poly}(d/\varepsilon)}{\eta}$

2: Set initialization $\mathbf{W}_{\ell,j} \leftarrow \mathbf{W}^{(0)}_{\ell,j}$ and $V^D_{\ell,j} \leftarrow 0$.

3: Apply SGDA (Algorithm 4) with $N$ samples, learning rate $\eta$ for $T$ steps on the following GAN objective (with $c$ being a small constant such as 0.001):

$$\min_{\mathbf{W}_{\ell,j}} \max_{V^D_{\ell,j}} \left( \left( \mathbb{E}[D^{(1)}_{\ell,j}(X^\star_\ell)] - \mathbb{E}[D^{(1)}_{\ell,j}(X_\ell)] \right) - \sum_{r \in [m_\ell]} \|V^D_{\ell,j,r}\|_2^{1+c} \right)$$

$\diamond$    $\|V^D_{\ell,j,r}\|_2^{1+c}$ *is an analog of the weight*

4: $[\mathbf{W}_{\ell,j}]_p \leftarrow [\mathbf{W}_{\ell,j}]_p / \|[\mathbf{W}_{\ell,j}]_p\|_2$

---

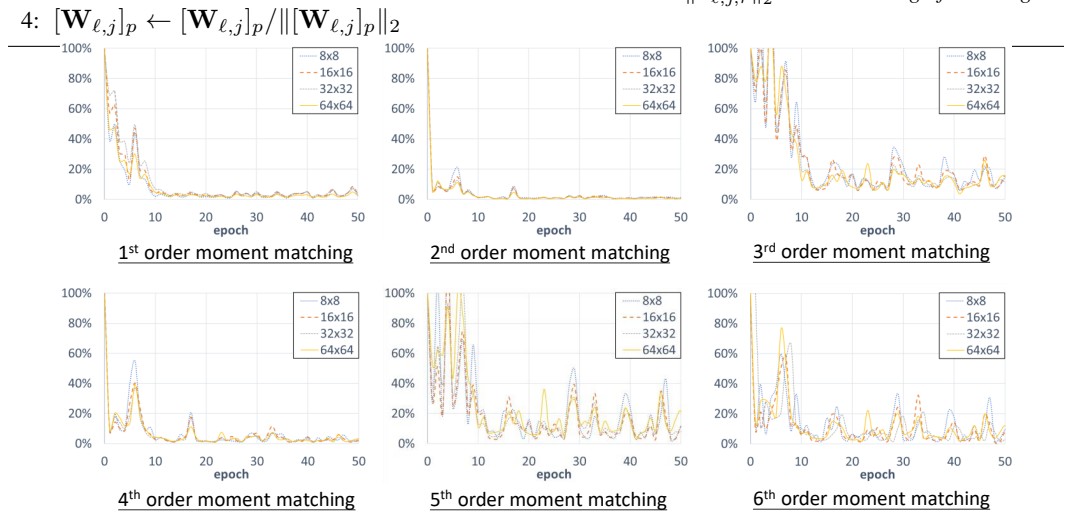

Figure 4: Difference between the moments of a generator's output and the true distribution. The $x$-axis is the number of epochs and the $y$-axis quantifies how close the moments are (the smaller the closer). Details are in Figure 9. One can see that the moments begin to match after epoch 10.

*moment matching within each patch* even at the earlier stage of training.

## 3.2   LEARN THE FIRST HIDDEN LAYER

Due to space limitation we defer the pseudocode and algorithm details of GAN_FirstHidden to the full version of this paper. However, we give the high level intuitions below.

HIGH-LEVEL INTUITIONS.   In the process of learning the lowest-resolution images $X^\star_1$, one cannot hope for (even approximately) learning the exact matrices $\mathbf{V}^\star_{1,j}$, or the exact function that maps from $z \mapsto X^\star_1$ (because $z$ is unknown during the training). Instead, the task is for learning the *distribution* of $X^\star_{1,j} = \mathbf{W}^\star_{1,j}\mathsf{ReLU}(\mathbf{V}^\star_{1,j}z - b^\star_{1,j})$.

Suppose for a moment that $\mathbf{W}^\star_{1,j}$ are already fully learned; then, it is perhaps not surprising that for the remaining part $\mathcal{S}^\star_{1,j} = \mathsf{ReLU}(\mathbf{V}^\star_{1,j}z - b^\star_{1,j})$, if we can somehow

1. learn the marginal distribution of $[\mathcal{S}^\star_{1,j}]_r$ for each $j, r$, and

2. learn the joint distribution of $\left( [\mathcal{S}^\star_{1,j}]_r, [\mathcal{S}^\star_{1,j'}]_{r'} \right)$ for each pair $(j,r) \neq (j',r')$,

then, we can recover the joint distribution of $\{[\mathcal{S}^\star_{1,j}]_r\}_{j,r}$. (As an analogy, for joint Gaussian, it suffices to learn the pair-wise correlation.) To achieve this, we design discriminators $D^{(4)}$ and $D^{(5)}$.

- $D^{(4)}$ discriminates the mismatch from single neurons by ensuring[5]

$$\mathbb{E}\,\widetilde{\mathsf{ReLU}}\left([(\mathbf{W}^D_{1,j})^\top X_{1,j}]_r - b\right) \approx \mathbb{E}\,\widetilde{\mathsf{ReLU}}\left([(\mathbf{W}^D_{1,j})^\top X^\star_{1,j}]_r - b\right)$$

---

[5]Like in the previous subsection, we shall periodically set $\mathbf{W}^D_{\ell,j} = \mathbf{W}_{\ell,j}$ to be the same as the generator from a previous check point; and the bias $b \ll 1$.

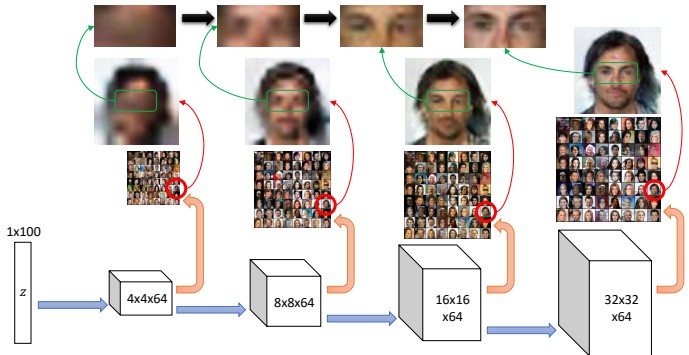

Figure 5: Forward super-resolution is a local operation; more details in Figure 7.

$$\mathbb{E}\,\widetilde{\mathsf{ReLU}}'\left([(\mathbf{W}_{1,j}^D)^\top X_{1,j}]_r - b\right) \approx \mathbb{E}\,\widetilde{\mathsf{ReLU}}'\left([(\mathbf{W}_{1,j}^D)^\top X_{1,j}^\star]_r - b\right)$$

Furthermore, as long as $\mathbf{W}_{1,j}^D$ is moderately learned, the sparse coding structure shall ensure $(\mathbf{W}_{1,j}^D)^\top X_{1,j} \approx \mathcal{S}_{1,j}$ and $(\mathbf{W}_{1,j}^D)^\top X_{1,j}^\star \approx \mathcal{S}_{1,j}^\star$. For such reason, and using $b \ll 1$, applying gradient descent ascent using discriminator $D^{(4)}$, in fact guarantees

$$\mathbb{E}\,\widetilde{\mathsf{ReLU}}\left([\mathcal{S}_{1,j}]_r\right) \approx \mathbb{E}\,\widetilde{\mathsf{ReLU}}\left([\mathcal{S}_{1,j}^\star]_r\right) \quad \text{and} \quad \mathbb{E}\,\widetilde{\mathsf{ReLU}}'\left([\mathcal{S}_{1,j}]_r\right) \approx \mathbb{E}\,\widetilde{\mathsf{ReLU}}'\left([\mathcal{S}_{1,j}^\star]_r\right)$$

Recall $[\mathcal{S}_{1,j}^\star]_r$ behaves as $\mathsf{ReLU}(g)$ for $g \sim \mathcal{N}(-\mu, \sigma^2)$ and has only 2 degrees of freedom; thus, matching moments on $\widetilde{\mathsf{ReLU}}$ and $\widetilde{\mathsf{ReLU}}'$ can learn the distribution of a single neuron $[\mathcal{S}_{1,j}^\star]_r$.

- $D^{(5)}$ discriminates the mismatch from the moments across two neurons, by ensuring

$$\mathbb{E}\left[\widetilde{\mathsf{ReLU}}\left([(\mathbf{W}_{1,j}^D)^\top X_{1,j}]_r - b\right)\widetilde{\mathsf{ReLU}}\left([(\mathbf{W}_{1,j'}^D)^\top X_{1,j'}]_{r'} - b\right)\right]$$
$$\approx \mathbb{E}\left[\widetilde{\mathsf{ReLU}}\left([(\mathbf{W}_{1,j}^D)^\top X_{1,j}^\star]_r - b\right)\widetilde{\mathsf{ReLU}}\left([(\mathbf{W}_{1,j'}^D)^\top X_{1,j'}^\star]_{r'} - b\right)\right]$$

For similar reason, gradient descent ascent learns to match moments on the cross terms:

$$\mathbb{E}\,\widetilde{\mathsf{ReLU}}\left([\mathcal{S}_{1,j}]_r\right)\widetilde{\mathsf{ReLU}}\left([\mathcal{S}_{1,j'}]_{r'}\right) \approx \mathbb{E}\,\widetilde{\mathsf{ReLU}}\left([\mathcal{S}_{1,j}^\star]_r\right)\widetilde{\mathsf{ReLU}}\left([\mathcal{S}_{1,j'}^\star]_{r'}\right)$$

We show this corresponds to learning $\langle[\mathbf{V}_{1,j}^\star]_r, [\mathbf{V}_{1,j'}^\star]_{r'}\rangle$ to a moderate accuracy.

In sum, if we apply SGDA on $D^{(4)}$ and $D^{(5)}$ together, we can hope for learning $\mathbf{V}_1$ up to a unitary transformation (see Lemma I.18). This ensures that we learn the distribution of $X_1^\star$.

## 3.3 LEARN HIGHER HIDDEN LAYERS

For resolution $\ell > 1$, patch $j \in [d_\ell]$, channel $r \in [m_\ell]$, to learn $[\mathbf{V}_{\ell,j}^\star]_r$, we introduce discriminator $D_{\ell,j,r}^{(2)}(Y_1, Y_2)$. It takes as input images of two resolutions: one should think of either $(Y_1, Y_2) = (X_\ell^\star, X_{\ell-1}^\star)$ comes from the true distribution, or $(Y_1, Y_2) = (X_\ell, X_{\ell-1})$ from the generator.

$$D_{\ell,j,r}^{(2)}(Y_1, Y_2) := \widetilde{\mathsf{abs}}\left(s_r - \mathsf{LeakyReLU}(\mathfrak{s}_r)\right)$$
$$\text{where} \quad \widetilde{\mathsf{abs}}(x) := \widetilde{\mathsf{ReLU}}(x - b) + \widetilde{\mathsf{ReLU}}(-x - b)$$
$$s_r := \left[\left([\mathbf{W}_{\ell,j}^D]^\top Y_{1,j}\right)\right]_r$$
$$\mathfrak{s}_r := \left(\sum_{j' \in \mathcal{P}_{\ell,j}} \mathbf{V}_{\ell,j,j'}^D \mathsf{LeakyReLU}\left([\mathbf{W}_{\ell-1,j'}^D]^\top Y_{2,j'}\right) - b_{\ell,j}^D\right)_r$$

Above, again $\mathbf{W}_{\ell,j}^D, \{\mathbf{W}_{\ell-1,j'}^D\}_{j' \in [d_{\ell-1}]}, b$ are default parameters (changed only periodically).

On the discriminator side, $\{[\mathbf{V}_{\ell,j,j'}^D]_r\}_{j' \in \mathcal{P}_{\ell,j}}, [b_{\ell,j}^D]_r$ are the actual trainable parameters; on the generator side, $\{[\mathbf{V}_{\ell,j,j'}]_r\}_{j' \in \mathcal{P}_{\ell,j}}, [b_{\ell,j}]_r$ as the trainable parameters. We note this discriminator $D^{(2)}$ is a **three-hidden layer neural network**. Yet, we show that such an network (together with the generator) can still be trained efficiently using gradient descent ascent.

**Algorithm 2** (GAN_FowardSuperResolution) using super-resolution to learn higher hidden layers

---

**Input:** $\mathbf{W}_\ell^{(0)}, \mathbf{W}_{\ell-1}^{(0)}, b, \ell, j$

1: Set default parameters $\mathbf{W}_{\ell,j}^D \leftarrow \mathbf{W}_{\ell,j}^{(0)}, \mathbf{W}_{\ell-1,j'}^D \leftarrow \mathbf{W}_{\ell-1,j'}^{(0)}$;
2: $N \leftarrow \frac{1}{\mathsf{poly}(d/\varepsilon)}, \eta \leftarrow \frac{1}{\mathsf{poly}(d/\varepsilon)}, T \leftarrow \frac{\mathsf{poly}(d/\varepsilon)}{\eta}; \lambda_G, \lambda_D \leftarrow \frac{1}{\mathsf{poly}(d/\varepsilon)}$
3: Initialize $\mathbf{V}_{\ell,j,j'} = \mathbf{V}_{\ell,j,j'}^D = \mathbf{I}$ for one of $j' \in \mathcal{P}_{\ell,j}$ and setting others as zero. Initialize $b_{\ell,j} = 0$.
4: **for** $r \in [m_\ell]$ **do**
5:     Apply SGDA with $N$ samples, learning rate $\eta$ for $T$ steps on the following GAN objective

$$\min_{\{[\mathbf{V}_{\ell,j,j'}^D]_r\}_{j' \in \mathcal{P}_{\ell,j}}, [b_{\ell,j}^D]_r;} \max_{\{[\mathbf{V}_{\ell,j,j'}]_r\}_{j' \in \mathcal{P}_{\ell,j}}, [b_{\ell,j}]_r} \left( \mathbb{E}[D_{\ell,j,r}^{(2)}(X_\ell^\star, X_{\ell-1}^\star)] - \mathbb{E}[D_{\ell,j,r}^{(2)}(X_\ell, X_{\ell-1})] \right)$$

$$- \lambda_G \|\mathbf{V}_\ell\|_F^2 + \lambda_D \|\mathbf{V}_\ell^D\|_F^2$$

6:     $[b_{1,j}]_r \leftarrow [b_{1,j}]_r + \mathsf{poly}(k_1)b$.

---

INTUITION: WHAT DOES THE DISCRIMINATOR DO? In this case, applying gradient descent ascent on $D^{(2)}$ actually learns *how to "super-resolute" the image from resolution level $\ell - 1$ to level $\ell$.* In particular, the discriminator wants to find a way where the patches $(X_{\ell,j}, X_{\ell-1,j'})$ differ statistically from the patches $(X_{\ell,j}^\star, X_{\ell-1,j'}^\star)$. For example, it can discriminate when $X_{\ell-1,j'}^\star = v_1 \implies X_{\ell,j}^\star = v_2$, but $X_{\ell-1,j'} = v_1, X_{\ell,j} \neq v_2$. In essence, it is discriminating the way where the generator super-resolutes a patch $X_{\ell,j}^\star$ from lower resolution *differently* from that of the true distribution.

As we demonstrate in Figure 5, such "super-resolution" operation is local, meaning that the learning process can be *separated* to learning over *individual patches*. The global structure across different patches of the images are learned in lower resolutions. This makes the learning process much simpler comparing to learning the full image from scratch. [6] We also provide empirical justification of the power of this "forward super-resolution", as in Figure 10(top) of the full paper: higher layers can indeed learn to super-resolute from the lower resolution images, which makes the learning much easier comparing to learning from scratch.

### 3.4 FINAL ALGORITHM

We implement our full algorithm in Algorithm 6 (see full paper). It performs layer-wise training. In each outer loop $\ell = 1, 2, \ldots, L$, it first warm-starts the output layer $\{\mathbf{W}_{\ell,j}\}_{j \in [d_\ell]}$ — note those weights are still very inaccurate.[7] Next, for this layer $\ell$, Algorithm 6 alternatively:

- uses the current output layer $\mathbf{W}_{\ell,j}$ to learn the hidden variables $\mathbf{S}_{\ell,j}$ (or equivalently the weights $\mathbf{V}_{\ell,j}, b_{\ell,j}$) to some accuracy — by applying GAN_FirstHidden if $\ell = 1$ or GAN_FowardSuperResolution if $\ell \geq 2$; and
- uses the current hidden variables $\mathbf{S}_{\ell,j}$ to learn the output layer $\mathbf{W}_{\ell,j}$ to an even better accuracy — by applying GAN_OutputLayer.

This alternating process repeats for $T' = \widetilde{O}(1)$ stages. Once again, we have broken the learning into multiple parts for analysis purpose, so it becomes clear how the generator can leverage the discriminator at different stages to learn the target distribution. (With more careful choices of learning rates, one can also combine them altogether.) Please note besides a simple SVD warm-start that is called only once per output layer $\mathbf{W}_{\ell,j}$, all the learning is done using minmax optimization on a generator-discriminator objective.

**What's in Full Paper.** We encourage readers to see our full paper at https://arxiv.org/abs/2106.02619. In the full version, we includes more related works and missing figures to better support the connection between our theory and practice. We also includes missing details for our technical assumptions from Section 2, and pseudocodes from Section 3. We restate our main theorem and ***the high level proof plan***, and shall also discuss limitations and open directions there.

---

[6] At resolution 1 the learning is global; in this case the one-hidden-layer generator can be trained via SGDA to capture the "global structure" of images (see Section 3.2 and Figure 1), with the help from properties of Gaussian random variable.

[7] Since the hidden variables $\mathbf{S}_{\ell,j}$ at this layer $\ell$ — which depend on weights $\{\mathbf{V}_{\ell,j}\}_{j \in [d_\ell]}$ — are still *not learned*, at this point, the best one can do is to look at the data covariance and give $\mathbf{W}_{\ell,j}$ a very rough estimate.

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
