# OpenReview forum: "Forward Super-Resolution: How Can GANs Learn Hierarchical Generative Models for Real-World Distributions"
_ICLR.cc/2023/Conference — ICLR 2023 poster_

### Official Review · Reviewer_okYU · 2022-10-24

**Confidence:** 2
**Correctness:** 4
**Technical Novelty And Significance:** 4
**Empirical Novelty And Significance:** 3
**Recommendation:** 6

**Clarity, Quality, Novelty And Reproducibility:**

Novelty: if the papers analysis holds, it will be the first theoretical treatment of GANs to my knowledge
Quality: Presentation and arguments are well done as far as I could check

Reproducibility/Clarity: I don't think the tight schedule of ICLR is a good fit for reviewing this work and the clarity suffers by having to be divided between hints and motivation in the main body and the appendix, despite both of them being well done. Overall, both are acceptable in my eyes since most important assumptions as well as notation and results are "seeded" in the main body.

**Strength And Weaknesses:**

Strengths of the paper:

- while the appendix is quite overwhelming in size, it *is* quite clearly written and legible. Overall, I applaud the authors for attempting to make such an involved work readable within the constraints of ICLR
- I could not verify the proofs in detail, but they seem reasonable to me as far as I could check. If it holds, the work could be the opening gate to a theory of GANs (albeit a very specific type which reminds me a lot of diffusion models in small local changes over multiple layers and relying on supervision in what will be latent stages)


Weaknesses

- the paper is very ambitious with just how much it is trying to present and I'm not sure it is the best way of doing things. A journal submission might have been a better fit, alternatively putting more emphasis on the interpretation and leaning *even more* on the already exhaustive appendix might have been a better move. However, it is difficult to thread this needle, and despite the reviewer guidance I cannot give actionable feedback on *how* to do this rewrite.

**Summary Of The Paper:**

The paper provides a theoretical model of *how* GANs might learn to generate images. The paper defers most of the work to the 71 page appendix, but if I understood it correctly, they model GAN training as iteratively learning "linear" (as in generalised linear model,i.e. including a single nonlinearity ala logistic regression) discriminator/generator layers on objectives at the given resolution, assuming that such a generator can generate the data (with additional sparse basis, non concentration and a couple of other assumptions).

The main theorem is a high probability statement about the trained generator being able to match the true generator (up to a transformation by a column orthonormal basis in the latent state) with high probability.

**Summary Of The Review:**

I applaud the authors for what attempting to fit what appears to be a very good 80 page paper into a 13 page conference format, but I am unsure whether it was a good idea. I lean towards a marginal accept because I could not do the required level of verification to stand behind the results stronger and I find a paper which relies almost exclusively on the appendix for verification to be problematic. Again, a journal venue or conference with different constraints might have fit better. However, what I could verify seems correct and I understand the challenge of presenting a work as involved  as this. Hence, marginal accept.

---

> ### Author Response · Authors · 2022-11-19
> **Response to Reviewer okYU**
>
> We really wish to thank the reviewer for appreciating our efforts to make the appendix “clearly written and legible.” We are also grateful that the reviewer agrees that our work “could be the opening gate to a theory of GANs.”
>
> ----
>
> We hear from the reviewer’s point that the length of the paper may suggest that we should submit it to a journal. We still, sincerely hope, that there could be a chance that this paper can first appear in a top conference  before we consider submitting its longer version to a journal, as the field of CS is very conference-driven.
>
> We believe our paper is one of the few that considers the theoretical foundations of image-like distribution generation, and deserves to be presented to the general ML community. It would be a pity if such an original, novel work misses the opportunity to be presented at conferences. We thus would like to sincerely ask for your reconsideration of the rating (as 6-6-6 may not be enough for acceptance?)
>
> ----
>
> As for the reviewer’s concerns about the scientific correctness of this paper, we’d like to mention that we have been distributing this paper and giving talks about it for more than a year, and so far there is no objection to the technical correctness of our result.
>
> ----
>
> Lastly, we find it very interesting that this paper also reminds the reviewer of diffusion models. In a long run, we would hope that our theory could guide us and help us better understand and design DL models and the training process. As a wide guess, the forward super-resolution hypothesis may help us understand generative models for languages — in the decoder part of a transformer, can the hidden features at different levels generate the desired words but at different “resolutions”? Like to first generate a bag of words, then a sub-bag of words, etc. As another wide guess, one might want to connect our theory to diffusion models, which also studies some form of super resolution but is on the pixel levels. We are learning diffusion models and hope to create something there.
>
> Thanks again!

---

### Official Review · Reviewer_EgRr · 2022-10-25

**Confidence:** 2
**Correctness:** 4
**Technical Novelty And Significance:** 3
**Empirical Novelty And Significance:** Not applicable
**Recommendation:** 6

**Clarity, Quality, Novelty And Reproducibility:**

* The main text is well-written and authors have provided intuitions for various choices and results. I did not check the proofs/appendix.
* The analysis is novel to the best of my knowledge but, given the disparity from practical GAN implementations, I am not sure how significant/useful the results are for the ICLR community.
* Reproducibility: N/A, theory work.

**Strength And Weaknesses:**

[Strengths]
* The paper presents an interesting analysis of a version of GANs.
* The paper is well-written (given the space limit) and the authors have made attempts to give intuitions and visuals to explain the results.

[Weaknesses]
* The analysis is done on a model/network that is not close to practice. Specifically, the discriminator is awkwardly split up into 3 parts which is not seen in practice, as the authors have themselves acknowledged. Some kind of empirical justification for the proposed learning algorithm is also missing.
* Overall, it seems like a conference paper (with a page limit) is not enough to properly explains the ideas presented in this work. The frequent references to the appendix break the reading flow. Therefore, in my subjective opinion, this work is more suitable for a longer format journal.

**Summary Of The Paper:**

The paper presents a theoretical analysis of GANs and shows how GANs can learn hierarchically generated distributions. Here hierarchical generation means that the distribution comes from an unknown generator that follows a forward super-resolution structure where each successive layer gives a higher resolution image (somewhat similar to progressively growing GANs). It is further assumed the learned generator also follows a similar structure. A learning algorithm is then presented using 3 types of discriminators (for the output layers, for the first hidden layers, and for remaining hidden layers) and it is shown that the generator trained using this algorithm learns the target distribution.

**Summary Of The Review:**

Overall, this paper presents an interesting theoretical analysis of GANs. However, there is a clear disparity between the model and learning algorithm studied in this paper vs. what's used in practice. Therefore, I am unsure about the significance of the results. It also appears that 9 pages are too few to properly explain the ideas presented in this work and a long-form journal submission may be a better choice.

Note: I do not have background in learning theory, so I am unable to provide an in-depth review of the analysis and proofs.

---

> ### Author Response · Authors · 2022-11-19
> **Response to Reviewer EgRr**
>
> We thank the reviewer for the time and agreeing that our “analysis is novel”. We hear that your main concern is a disparity between the model and learning algorithm studied in this paper vs. what's used in practice.
>
> We’d like to defend ourselves and provide perhaps a bigger context. At the end of the day, we wish to have theories for deep learning (DL) because some of them can help us better understand and design DL models, training processes as opposed to being somewhat blind. As a wide guess, the forward super-resolution hypothesis may help us understand generative models for languages — in the decoder part of a transformer, can the hidden features at different levels generate the desired words but at different “resolutions”? Like to first generate a bag of words, then a sub-bag of words, etc. As another wide guess, one might want to connect our theory to diffusion models, which also studies some form of super resolution but is on the pixel levels.
>
> Depending on the task, however, getting theory done for DL may be either hard or easy. Recall if the task is as simple as to prove for efficient training convergence, then one can look at ReLU and study neural tangent kernel, dating back to 1811.03962. But to prove a learnability result like us, it is significantly more difficult. Even in the case of a feedforward NN, hardly any theory result can go beyond two or three layers (except making strong assumptions such as looking at polynomial networks). In GAN learning, having a complex generator together with a complex discriminator, it may be really beyond our reach at this moment.
>
> It is for such a reason, we have simplified and designed some artificial, smaller discriminators. This is our biggest difference to practical GAN training, but we view this as an approximation of the real-world algorithm by breaking it into more interpretable, and even provable pieces. Indeed, we verified that the functionalities of those smaller discriminators are indeed subsumed by the large one used in practice --- thus we believe our theory can explain how those (hierarchically generated) images are learned in real-life GAN training as well.
>
> As for other disparities, we believe all of them are rather minor. For instance, we used layerwise training but Figure 10 shows that’s okay. We added an output layer for each resolution level, but that may be even desirable according to ProgressiveGAN in Karras et al. (2018). We also used smoothed ReLU, a warmup, etc, but those are quite standard in theory literature to simplify proofs — they are truly not needed in practice.
>
> ----
>
> We noticed that the reviewer ranked our paper as “Aspects of the contributions exist in prior work”. We are not sure what aspect of our theory contributions can be considered to "have existed in prior works"? We believe prior theory works largely fail to address the learnability problem of GANs (i.e., on how those generative models are learned), and please see pages 1-2.
>
> ---
>
> Finally, we hear from the reviewer’s point that our length of the paper may suggest us submitting to journals. We still, sincerely hope, that there could be a chance that this paper can first appear in a top conference before we consider submitting its longer version to a journal.
>
> We believe our paper is one of the few that considers the theoretical foundations of image-like distribution generation, and deserves to be presented to the general ML community. It would be a pity if such an original, novel work misses the opportunity to be presented at conferences.
>
> We thus sincerely hope the reviewer can re-consider your rating. Thanks!

---

### Official Review · Reviewer_2F3d · 2022-10-26

**Confidence:** 3
**Correctness:** 3
**Technical Novelty And Significance:** 3
**Empirical Novelty And Significance:** 3
**Recommendation:** 6

**Clarity, Quality, Novelty And Reproducibility:**

Novel in idea and theoretical analysis. However, it is not easy to reproduces the empirical analysis as not all details are provided. The theory is overwhelming (e.g., in supplementary) but the authors done well by trying to simplify it and with some high-level intuitions in main papers.

**Strength And Weaknesses:**

Strengths:

1. The theory about the “forward super-resolution” is new to me.

2. The few proofs in the paper looks correct, but most of them cannot be verified due to the limited time.

3. The new learning algorithm for GAN with the theoretical supports.

Weaknesses:

1. The paper aims to discuss assumption “forward super-resolution” holds in practice and show the proposed underlying training mechanism simulates the actual learning process of GANs on real-world problems. However, excepts some examples which was developed in some existing GANs, I cannot find anywhere in the paper. Can the author point to where this is discussed? Seems to me if most of them only with discussion rather than theoretical or empirical evidence which is not convincing to me enough.

2. Moreover, can the authors details how to design the empirical experiments for Figs 2, 3, 4?  In Fig. 3, it is not convincing to me that the learned of network is really sparse (it may be depends on how we design sparsity) but to me it is more on the long-tail distribution since most of columns have the value. Also, the sparsity assumption may depend on the network capacity and how much knowledge of the data needs to be encoded into the networks. For example, can the authors reduce the network at different sizes to see whether it still learn the sparse structure?

3. One suggestion is designing the toy examples to show the learning capability of proposed algorithm as claimed by the authors it is natural in real-world, designing such toy examples may not difficult?

**Summary Of The Paper:**

The paper proposes the distribution structure called “forward super-resolution” (forward super-resolution assumes that for image resolution at level $L$ is represented by $G_l*W_l$, where $G_l$ is hidden neuron at layer l of neural network $G$ (with RELU activation) and $W_l$ is certain matrix), which is hierarchically generated, which can be learnt effectively by GAN using stochastic gradient de- scent ascent (SGDA) under certain conditions. In particular, to learn the forward-super resolution, the authors assumes that the true distribution of layers in neural network G follows sparse structure and “not-too-positively correlated” at patch level. With this assumption, the authors provide the theoretical capability and the convergence of the generator and design the learning algorithms for it.  The algorithm breaks the learning into multiple parts (GAN OutputLayer, GAN FirstHidden, GAN FowardSuperResolution) and show how the training of each part is formed and converged. The main idea of the convergence theory showing the generator will learn to match the moments of the true distribution.

**Summary Of The Review:**

Overall, the high-level idea is interesting and novel to me. Indeed, empirically looks like learning via hierarchical approach in GAN seems more effective as shown in some existing works. However, this is too dense theoretical paper up to 71 pages, challenging for me to verify the correctness of the proofs in limited time. I have a few questions regarding the empirical evidence and main assumptions that hopefully the authors can clarify in the rebuttal.

---

> ### Author Response · Authors · 2022-11-19
> **Response (2/2) to Reviewer 2F3d**
>
> **Q**: Moreover, can the authors details how to design the empirical experiments for Figs 2, 3, 4?
>
> **A**: We train a standard 4-layer DCGAN using progressive training (i.e., forward super-resolution). That is, we added a (deconvolutional) output linear layer $W_\ell$ to the hidden neurons on each layer $\ell$, and used that to match the images at resolution level $\ell$.  Please note we did not regularize the activation to be sparse — they simply learned to be sparse automatically.
>
> ---
>
> **Q**:  In Fig. 3, it is not convincing to me that the learned of network is really sparse (it may be depends on how we design sparsity) but to me it is more on the long-tail distribution since most of columns have the value. Also, the sparsity assumption may depend on the network capacity and how much knowledge of the data needs to be encoded into the networks. For example, can the authors reduce the network at different sizes to see whether it still learn the sparse structure?
>
> **A**: First, yes, if one varies the sizes of the network, the sparsity and negative correlation properties also hold in practice — except if the network width is too small then learning images becomes somewhat unsuccessful. We varied the network width by a factor of 1/2, 2, 4 and 8, and similar distributions as Figure 3 + 6 also show up. Would you like to have us add this experiment in the next version? it will occupy a few pages just like Figure 6 though.
>
> And indeed for the simplicity of theory, we have assumed some absolute version of sparsity while in practice it is only approximately sparse. Our theory can actually support a bit of noise on the hidden neurons, but we refrained from proving that more complex case.
>
> ---
>
> **Q**:  One suggestion is designing the toy examples to show the learning capability of proposed algorithm as claimed by the authors it is natural in real-world, designing such toy examples may not difficult?
>
> **A**:  We thank the reviewer for the suggestion. As mentioned above, we proposed an algorithm as an approximation of the real-world GAN training algorithm by breaking it into more interpretable, and even provable pieces. We are not advocating for a new algorithm, and thus did not think of implementing it. Instead, we spent more space in the paper verifying that our algorithmic changes (for theoretical supposes) are reasonable — they are consistent with the practical learning process of GANs.
>
> ---
>
> **PS**: we noticed that the reviewer ranked our paper as “Aspects of the contributions exist in prior work”. We are not sure what aspect of our theory contributions can be considered to "have existed in prior works"? We believe prior works are very inadequate for addressing the learnability problem of GANs (i.e., on how those generative models are learned), and please see pages 1-2.
>
> We thus sincerely hope the reviewer can re-consider your rating. Thanks!

---

> ### Author Response · Authors · 2022-11-19
> **Response (1/2) to Reviewer 2F3d**
>
> We thank the reviewer for spending time on reading this paper! We provided below a detailed reply and sincerely hope the reviewer can reconsider your evaluation.
>
> Before we address the concerns, let us first emphasize that our submission is a DL theory paper that tries to explain how GANs work, as opposed to proposing new algorithms. We view our algorithm as an approximation of the real-world GAN training by breaking it into more interpretable, and even provable pieces.
>
> Recall it is usually impossible to prove what exactly happens in practical DL, so theoreticians are willing to make assumptions. Usually, the harder the theorem is, the more complex assumptions are needed.  If the task is as simple as to prove efficient training convergence, then one can simply look at ReLU and study neural tangent kernel, dating back to 1811.03962. We thus hope the reviewer can be reasonably forgiving us for making seemingly too many assumptions – because we are shooting for a higher bar.
>
> ---
>
> **Q**: The paper aims to discuss assumption “forward super-resolution” holds in practice and show the proposed underlying training mechanism simulates the actual learning process of GANs on real-world problems. However, excepts some examples which was developed in some existing GANs, I cannot find anywhere in the paper. Can the author point to where this is discussed? Seems to me if most of them only with discussion rather than theoretical or empirical evidence which is not convincing to me enough.
>
> **A**: Sorry if we did not explain this perfectly in the current version. Our formal notion of “forward super-resolution” can be found on page 2.  That is, letting $G_\ell$ represent the hidden neuron values at layer $\ell$, then the distribution of images at resolution ell is given by $W_\ell G_\ell$. We verified this empirically as early as in Figure 1. That is, for DCGAN, one can train $W_\ell$ using $G_\ell$ at four different layers $\ell=1,2,3,4$ to shoot for four different resolutions. Those $W_\ell$’s are deconvolutional and we present them in the middle of Figure 1 (as well as top row of Figure 2). This confirms that low-resolution images can indeed be learned via low-level hidden features.
>
> As for demonstrating that “the proposed underlying training mechanism simulates the actual learning process of GANs on real-world problems”, we emphasize that the main difference between our theory and practice, is that (1) we used layerwise training as opposed to training all the layers at once, (2) we added an output layer $W_\ell G_\ell$ at each resolution level $\ell$, and (3) we used some small, moment-matching-like discriminators. (We have other minor differences, like a warmup, smoothed ReLU, etc. But those are standard in theory literature to tradeoff for easier proofs and are less important.)
>
> As for differences (1)+(2), we verified in the top row of Figure 10 (on page 18 of the appendix) that layerwise training may not be too different from standard GAN training in practice. At least, it is much more realistic than neural kernel methods (e.g. NTK) used in most DL theories, as shown in the bottom row of Figure 10.
>
> As for difference (3), we verified in Figure 4 that real-life GANs are also performing moment matching. Indeed, we are not advocating for a new training algorithm. We simplified the discriminator used in practice for the sake of proving our main theorem, and then verified that the functionalities of those smaller discriminators are indeed subsumed by the larger one used in practice --- thus we believe our theory can explain how those (hierarchically generated) images are learned in real-life GANs

---

### Decision · Program_Chairs · 2023-01-20

**Decision:**

Accept: poster

**Justification For Why Not Higher Score:**

The paper's setup is clearly somewhat restrictive and thus does not merit a higher score.

**Justification For Why Not Lower Score:**

All reviewers suggested acceptance.

**Metareview: Summary, Strengths And Weaknesses:**

In this work, it is shown that generative adversarial networks (GANs) can efficiently learn certain hierarchically generated distributions that are close to the distribution of real-life images. It is proven that when a distribution has a structure referred to as "forward super-resolution," then training GANs using stochastic gradient descent ascent (SGDA) can learn this distribution efficiently in terms of sample and time complexities. Additionally, empirical evidence is provided to support the assumption of "forward super-resolution" and to demonstrate the underlying mechanisms that allow for the efficient training of GANs in practice.

The reviewers liked that the paper provided theory for “forward super-resolution” and thought the new learning algorithms for GANs that come with theoretical support are interesting. The reviewers also raised various concerns: (1) more theoretical/empirical evidence is needed for forward super-resolution (2) lack of detail on some of the experimental setup (3) dependence on sparsity level on the architecture. The authors rebuttal seemed to mitigate a lot of these concerns but the reviewers all kept their original marginal accept score. Overall the paper is interesting and merits publication. I therefore recommend acceptance.




**Note From Pc:**

if the above contains the word "oral" or "spotlight" please see: "oral" presentation means -> notable-top-5% and "spotlight" means -> notable-top-25%. As stated in our emails, we are disassociating presentation type from AC recommendations